# Improved Communication Efficiency in Federated Natural Policy Gradient via ADMM-based Gradient Updates

**Guangchen Lan**
Purdue University
West Lafayette, IN 47907
lan44@purdue.edu

**Han Wang**
Columbia University
New York, NY 10027
hw2786@columbia.edu

**James Anderson**
Columbia University
New York, NY 10027
anderson@ee.columbia.edu

**Christopher Brinton**
Purdue University
West Lafayette, IN 47907
cgb@purdue.edu

**Vaneet Aggarwal**
Purdue University
West Lafayette, IN 47907
vaneet@purdue.edu

## Abstract

Federated reinforcement learning (FedRL) enables agents to collaboratively train a global policy without sharing their individual data. However, high communication overhead remains a critical bottleneck, particularly for natural policy gradient (NPG) methods, which are second-order. To address this issue, we propose the FedNPG-ADMM framework, which leverages the alternating direction method of multipliers (ADMM) to approximate global NPG directions efficiently. We theoretically demonstrate that using ADMM-based gradient updates reduces communication complexity from $\mathcal{O}(d^2)$ to $\mathcal{O}(d)$ at each iteration, where $d$ is the number of model parameters. Furthermore, we show that achieving an $\epsilon$-error stationary convergence requires $\mathcal{O}(\frac{1}{(1-\gamma)^2\epsilon})$ iterations for discount factor $\gamma$, demonstrating that FedNPG-ADMM maintains the same convergence rate as the standard FedNPG. Through evaluation of the proposed algorithms in MuJoCo environments, we demonstrate that FedNPG-ADMM maintains the reward performance of standard FedNPG, and that its convergence rate improves when the number of federated agents increases.

## 1 Introduction

Policy gradient methods are commonly used to solve reinforcement learning (RL) problems in various applications, with recent popular examples being InstructGPT [30] and ChatGPT (GPT-4 [29]). Although first-order methods, such as Policy Gradient (PG) and Proximal Policy Optimization (PPO) [40], are favored for their simplicity, second-order methods, such as Natural Policy Gradient (NPG) [16, 36] and its practical version Trust Region Policy Optimization (TRPO) [38], are often seen to exhibit superior convergence behavior, e.g., on the Swimmer [40] and Humanoid [10] tasks in MuJoCo environments [44]. Furthermore, recent works have demonstrated that the convergence guarantees of NPG with Kullback–Leibler (KL) divergence constraints (second-order methods) are superior to those of PG (first-order methods) [21, 23], motivating closer inspection for practical use.

Many contemporary RL applications are large-scale and rely on high volumes of data for model training [20, 22, 33]. A conventional approach to enabling training on massive data is transmitting data collected locally by different agents to a central server, which can then be used for policy learning. However, this is not always feasible in real-world systems where communication bandwidth

37th Conference on Neural Information Processing Systems (NeurIPS 2023).

is limited and long delays are not acceptable [19, 28], such as in edge devices [19], Internet of Things [5, 27], autonomous driving [41], and vehicle transportation [3, 11]. Moreover, sharing individual data collected by agents raises privacy and legal issues [15, 26].

Federated learning (FL) [12, 25] offers a promising solution to the challenges posed by data centralization, where agents communicate locally trained models rather than raw datasets. However, FL is typically applied to supervised learning problems. Recent work has expanded the scope of FL to federated reinforcement learning (FedRL), where $N$ agents collaboratively learn a global strategy without sharing the trajectories they collected during agent-environment interaction [14, 17, 50]. Federated reinforcement learning has been studied in tabular RL [2, 14, 17], control tasks [47, 48] and for value-function based algorithms [46, 50], where linear speedup has been demonstrated. For policy-gradient based algorithms, we note that linear speedup is easy to see as the trajectories collected by each agent could be parallelized.

Efficient state-of-the-art guarantees for federated policy gradient based approaches can be achieved by Federated NPG (FedNPG), which is a second-order method. However, sharing second-order information increases the communication complexity, which is one of the fundamental challenges in FL [7, 42, 37]. In supervised FL, works including FedNL [37], BL [34], Newton-Star/Learn [13] and FedNew [7] have been recently proposed to reduce the communication complexity of second-order methods, typically by approximating Hessian matrices for convex or strongly convex problems. However, federated second-order *reinforcement learning* has not been investigated, which provides the key motivation for our work. The key question that this paper aims to address is:

***Can we reduce the communication complexity for second-order federated natural policy gradient approach while maintaining performance guarantees?***

We answer this question in the affirmative by introducing FedNPG-ADMM, an algorithm that estimates global NPG directions using alternating direction method of multipliers (ADMM) [8, 9, 45]. This estimation reduces the communication complexity from $\mathcal{O}(d^2)$ to $\mathcal{O}(d)$, where $d$ is the number of model parameters. However, it is non-trivial to see whether FedNPG-ADMM will maintain similar convergence guarantees as FedNPG. We show in this work that FedNPG-ADMM indeed does maintain these guarantees, and provides a speedup with the number of agents. The key contributions that we make are summarized as follows:

1. We propose a novel federated NPG algorithm, FedNPG-ADMM, where the global NPG directions are estimated through ADMM.

2. Using the ADMM-based global direction estimation, we demonstrated that the communication complexity reduces by $\mathcal{O}(d)$ as compared to transmitting the second-order information (standard FedNPG).

3. We prove the FedNPG-ADMM method achieves an $\epsilon$-error stationary convergence with $\mathcal{O}(\frac{1}{(1-\gamma)^2\epsilon})$ iterations for discount factor $\gamma$. Thus, it achieves the same convergence rate as the standard FedNPG.

4. Experimental evaluations in MuJoCo environments demonstrate that FedNPG-ADMM maintains the convergence performance of FedNPG. We also show improved performance as more federated agents engage in collecting trajectories.

## 2  Background

**Markov Decision Process:**  We consider the Markov decision process (MDP) as a tuple $\langle \mathcal{S}, \mathcal{A}, \mathcal{P}, \mathcal{R}, \gamma \rangle$, where $\mathcal{S}$ is the state space, $\mathcal{A}$ is a finite action space, $\mathcal{P} : \mathcal{S} \times \mathcal{A} \times \mathcal{S} \to \mathbb{R}$ is a Markov kernel that determines transition probabilities, $\mathcal{R} : \mathcal{S} \times \mathcal{A} \to \mathbb{R}$ is a reward function, and $\gamma \in (0, 1)$ is a discount factor. At each time step $t$, the agent executes an action $a_t \in \mathcal{A}$ from the current state $s_t \in \mathcal{S}$, following a stochastic policy $\pi$, i.e., $a_t \sim \pi(\cdot|s_t)$. For on policy $\pi$, a state value function is defined as

$$V_\pi(s) = \mathbb{E}_{\substack{a_t \sim \pi(\cdot|s_t), \\ s_{t+1} \sim P(\cdot|s_t, a_t)}} \left[ \sum_{t=0}^{\infty} \gamma^t r(s_t, a_t)|s_0 = s \right]. \tag{1}$$

Similarly, a state-action value function (Q-function) is defined as

$$Q_\pi(s,a) = \mathbb{E}_{\substack{a_t \sim \pi(\cdot|s_t), \\ s_{t+1} \sim P(\cdot|s_t, a_t)}} \left[ \sum_{t=0}^\infty \gamma^t r(s_t, a_t) | s_0 = s,\ a_0 = a \right]. \tag{2}$$

An advantage function is then define as $A_\pi(s,a) = Q_\pi(s,a) - V_\pi(s)$. With continuous states, the policy is parametrized by $\theta \in \mathbb{R}^d$, and then the policy is referred as $\pi_\theta$ (Deep RL parametrizes $\pi_\theta$ by deep neural networks). A state-action visitation measure induced by $\pi_\theta$ is given as

$$\nu_{\pi_\theta}(s,a) = (1-\gamma) \mathbb{E}_{s_0 \sim \rho} \left[ \sum_{t=0}^\infty \gamma^t P(s_t = s,\ a_t = a | s_0,\ \pi_\theta) \right], \tag{3}$$

where starting state $s_0$ is drawn from a distribution $\rho$. The *goal* of an agent is to maximize the expected discounted return defined as

$$J(\theta) = \mathbb{E}_{s \sim \rho} \left[ V_{\pi_\theta}(s) \right]. \tag{4}$$

The gradient of $J(\theta)$ can be written as [39]:

$$\nabla_\theta J(\theta) = \mathbb{E}_\tau \left[ \sum_{t=0}^\infty \left( \nabla_\theta \log \pi_\theta(a_t|s_t) \right) A_{\pi_\theta}(s_t, a_t) \right], \tag{5}$$

where $\tau = (s_0, a_0, s_1, a_1, \cdots)$ is a trajectory induced by policy $\pi_\theta$. We denote the policy gradient by $\mathbf{g}$ for short. In practice, we can sample $(s,a) \sim \nu^{\pi_{\theta^k}}$ and obtain the unbiased estimate $\widehat{A}_{\pi_{\theta^k}}(s,a)$ using Algorithm 3 in [1].

**Natural Policy Gradient (NPG):** At the $k$-th iteration, natural policy methods with a trust region [38] update policy parameters as follows

$$\theta^{k+1} = \arg\max_\theta \mathbb{E}_{s,a} \left[ \frac{\pi_\theta(a|s)}{\pi_{\theta^k}(a|s)} A_{\pi_{\theta^k}}(s,a) \right] \tag{6}$$
$$\text{s.t. } \overline{D}(\theta \| \theta^k) \le \delta.$$

where

$$\overline{D}(\theta \| \theta^k) = \mathbb{E}_s \left[ D\left( \pi_\theta(\cdot|s) \| \pi_{\theta^k}(\cdot|s) \right) \right], \tag{7}$$

$D(\cdot)$ is the KL-divergence operation, and $\delta > 0$ is the radius of the trust region. Practically, using the first-order Taylor expansion for the target value and the second-order Taylor expansion for the divergence constraint, (6) is expanded as follows

$$\theta^{k+1} = \arg\max_\theta \mathbf{g}^\top (\theta - \theta^k) \tag{8}$$
$$\text{s.t. } \frac{1}{2}(\theta - \theta^k)^\top \mathbf{H}(\theta - \theta^k) \le \delta,$$

where $\mathbf{H} = \nabla_\theta^2 \overline{D}(\theta \| \theta^k) \in \mathbb{R}^{d \times d}$, and the individual elements are given by $\mathbf{H}_{ij} = \frac{\partial}{\partial \theta_i} \frac{\partial}{\partial \theta_j} \mathbb{E}_s \left[ D\left( \pi_\theta(\cdot|s) \| \pi_{\theta^k}(\cdot|s) \right) \right] \Big|_{\theta = \theta^k}$. Using Lagrangian duality, the iterates of NPG ascent are expressed as

$$\theta^{k+1} = \theta^k + \sqrt{\frac{2\delta}{\mathbf{g}^\top \mathbf{H}^{-1} \mathbf{g}}} \mathbf{H}^{-1} \mathbf{g}. \tag{9}$$

**Federated NPG:** FedNPG is a paradigm in that $N$ agents collaboratively train a common global policy with parameters $\theta$ as illustrated in Figure 1 (a). During the training process, each agent computes gradients (and second-order matrices) using its *local data*; then, the gradients of all agents are transmitted to a central server. In particular, one FedNPG training iteration consists of the following three steps:

- Downlink Transmission: The server broadcasts the current global policy parameters $\theta \in \mathbb{R}^d$ to all $N$ agents.
- Uplink Transmission: Collecting its own local data $\mathcal{D}_i$ based on the common policy $\pi_\theta$, each agent $i$ computes its local gradient $\mathbf{g}_i \in \mathbb{R}^d$ and second-order matrix $\mathbf{H}_i \in \mathbb{R}^{d \times d}$. Then, it sends $\mathbf{g}_i$ and $\mathbf{H}_i$ back to the server.
- Global Update: The server averages local gradients and second-order matrices to get the global gradient and matrix as follows:

$$\mathbf{H} \leftarrow \frac{1}{N} \sum_{i=1}^{N} \mathbf{H}_i, \ \ \mathbf{g} \leftarrow \frac{1}{N} \sum_{i=1}^{N} \mathbf{g}_i. \tag{10}$$

The server then updates global policy parameters as

$$\theta \leftarrow \theta + \sqrt{\frac{2\delta}{\mathbf{g}^\top \mathbf{H}^{-1} \mathbf{g}}} \mathbf{H}^{-1} \mathbf{g}. \tag{11}$$

We use $|\mathcal{D}_i|$ to denote the size of collected data set $\mathcal{D}_i$. Without loss of generality, we take $|\mathcal{D}_1| = \cdots = |\mathcal{D}_N|$ for simplicity. As proven in [49], gradient update methods are *immune* to whether collected data is i.i.d., or not.

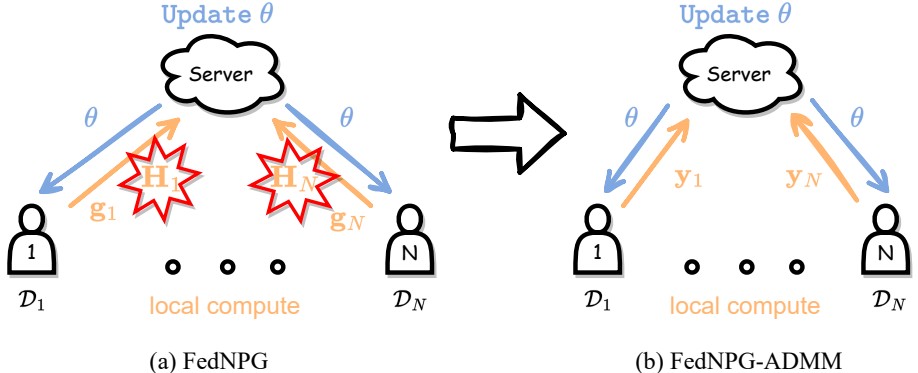

(a) FedNPG  (b) FedNPG-ADMM

Figure 1: An illustration of federated learning based on second-order methods with $N$ agents. (a) FedNPG via standard average. In the uplink, transmitting the matrix $\mathbf{H}_i$ brings $\mathcal{O}(d^2)$ communication complexity. (b) FedNPG-ADMM in this paper with only $\mathcal{O}(d)$ communication complexity.

**Bottleneck of Federated NPG:**  As shown in Figure 1, at each iteration, the server collects $\{\mathbf{H}_i \in \mathbb{R}^{d \times d}, \mathbf{g}_i \in \mathbb{R}^d\}_{i=1}^N$ from $N$ agents and updates global policy parameters as follows

$$\theta \leftarrow \theta + \sqrt{\frac{2N\delta}{(\sum_{i=1}^{N} \mathbf{g}_i)^\top (\sum_{i=1}^{N} \mathbf{H}_i)^{-1} \sum_{i=1}^{N} \mathbf{g}_i}} (\sum_{i=1}^{N} \mathbf{H}_i)^{-1} \sum_{i=1}^{N} \mathbf{g}_i. \tag{12}$$

We call this method a standard average FedNPG. The uplink communication complexity from each agent is $\mathcal{O}(d^2)$ in each iteration round. As the uplink is highly limited [43], applying (12) is not practical when $d$ is large.

In the next section, we introduce our ADMM-based approach as shown in Figure 1 (b), which reduces the communication complexity to $\mathcal{O}(d)$ at each iteration and meanwhile keeps convergence performances.

## 3  FedNPG via ADMM

To minimize the communication overhead in each round of communication, we begin by formulating a quadratic problem. The solution to this problem provides the updating direction in (12), as follows:

$$\left(\sum_{i=1}^{N}\mathbf{H}_i\right)^{-1}\sum_{i=1}^{N}\mathbf{g}_i = \arg\min_{\mathbf{y}} \frac{1}{2}\mathbf{y}^\top\left(\sum_{i=1}^{N}\mathbf{H}_i\right)\mathbf{y} - \mathbf{y}^\top\sum_{i=1}^{N}\mathbf{g}_i. \tag{13}$$

This minimization problem is equivalent to

$$\min_{\mathbf{y},\{\mathbf{y}_i\}_{i=1}^N} \sum_{i=1}^{N}\left(\frac{1}{2}\mathbf{y}_i^\top\mathbf{H}_i\mathbf{y}_i - \mathbf{y}_i^\top\mathbf{g}_i + \frac{\rho}{2}\|\mathbf{y}_i-\mathbf{y}\|^2\right) \tag{14}$$
$$\text{s.t. } \mathbf{y}=\mathbf{y}_i, \quad i=1,\cdots,N,$$

where $\rho > 0$ is a penalty constant and $\|\cdot\|$ denotes the Euclidean norm. This equivalence transforms the original quadratic problem into a distributed manner and is the key process of efficient FedNPG.

**Remark**: This approach does not aim to approximate the Hessian from each agent, but approximates the global direction $(\sum_{i=1}^{N}\mathbf{H}_i)^{-1}\sum_{i=1}^{N}\mathbf{g}_i$ directly. Note that this differentiates our approach from the work mentioned in the introduction.

The Lagrangian function associated to optimization problem (14) is

$$\mathcal{L}(\mathbf{y},\{\mathbf{y}_i\}_{i=1}^N,\{\lambda_i\}_{i=1}^N) = \sum_{i=1}^{N}\left(\frac{1}{2}\mathbf{y}_i^\top\mathbf{H}_i\mathbf{y}_i - \mathbf{y}_i^\top\mathbf{g}_i + \frac{\rho}{2}\|\mathbf{y}_i-\mathbf{y}\|^2 + \langle\lambda_i,\mathbf{y}_i-\mathbf{y}\rangle\right), \tag{15}$$

where $\{\lambda_i \in \mathbb{R}^d\}_{i=1}^N$ are dual variables. Next, we solve the distributed optimization problem through the alternating direction method of multipliers (ADMM) [4]. The policy update of FedNPG via one-step ADMM is given in Algo. 1.

---

**Algorithm 1** FedNPG-ADMM

---

**Input:** MDP $\langle\mathcal{S},\mathcal{A},\mathcal{P},\mathcal{R},\gamma\rangle$; Number of timesteps $T$; Penalty constant $\rho$; Step size $\eta$; Initial $\theta_0 \in \mathbb{R}^d$, $\mathbf{y}^0 \in \mathbb{R}^d$, $\{\mathbf{y}_i^0 = \mathbf{y}^0\}_{i=1}^N$, $\{\lambda_i \in \mathbb{R}^d\}_{i=1}^N$.
1: **for** $k=1,\cdots,K$ **do**
2:    ▷ Server broadcast
3:    Broadcast $\mathbf{y}^{k-1}$ and $\theta^{k-1}$ to $N$ agents.
4:    ▷ Agent update
5:    **for** each agent $i \in \{N\}$ **do in parallel**
6:       $\lambda_i \leftarrow \lambda_i + \rho(\mathbf{y}_i^{k-1} - \mathbf{y}^{k-1})$
7:       $\mathbf{g}_i^k \leftarrow \frac{1}{|\mathcal{D}_i|}\sum_{\tau\in\mathcal{D}_i}\sum_{t=0}^{T}\left(\nabla_{\theta^{k-1}}\log\pi_{\theta^{k-1}}(a_t|s_t)\right)\widehat{A}_{\pi_{\theta^{k-1}}}(s_t,a_t)$
8:       $\mathbf{y}_i^k \leftarrow (\mathbf{H}_i^k+\rho\mathbf{I})^{-1}(\mathbf{g}_i^k-\lambda_i+\rho\mathbf{y}^{k-1})$
9:       Transmit $\mathbf{y}_i^k \in \mathbb{R}^d$ and $\mathbf{g}_i^k \in \mathbb{R}^d$ to the server.
10:   **end for**
11:   ▷ Server update
12:   $\mathbf{y}^k \leftarrow \frac{1}{N}\sum_{i=1}^{N}\mathbf{y}_i^k$
13:   $\theta^k \leftarrow \theta^{k-1} + \eta\sqrt{\frac{2N\delta}{(\sum_{i=1}^{N}\mathbf{g}_i^k)^\top\mathbf{y}^k}}\cdot\mathbf{y}^k$
14: **end for**
**Output:** $\theta^K$

---

Agent $i$ computes $\mathbf{H}_i$ and $\mathbf{g}_i$ based on locally collected data. At each step of ADMM, agent $i$ updates $\mathbf{y}_i$ in line 8 as follows

$$\mathbf{y}_i = \arg\min_{\mathbf{y}_i}\left(\frac{1}{2}\mathbf{y}_i^\top\mathbf{H}_i\mathbf{y}_i - \mathbf{y}_i^\top\mathbf{g}_i + \frac{\rho}{2}\|\mathbf{y}_i-\mathbf{y}+\frac{\lambda_i}{\rho}\|^2\right)$$
$$\overset{8:}{=} (\mathbf{H}_i+\rho\mathbf{I})^{-1}(\mathbf{g}_i-\lambda_i+\rho\mathbf{y}), \tag{16}$$

where $\mathbf{I} \in \mathbb{R}^{d\times d}$ is the identity matrix. In practical implementation, conjugate gradient methods can be used to compute $\mathbf{y}_i$ for efficiency (Appendix C in [38]).

In line 12, after receiving $\mathbf{y}_i$ and $\mathbf{g}_i$ from all $N$ agents, the server updates the global search direction as follows

$$
\begin{aligned}
\mathbf{y} &= \arg\min_{\mathbf{y}} \sum_{i=1}^{N} \left( \frac{\rho}{2} \|\mathbf{y}_i - \mathbf{y}\|^2 + \langle \lambda_i, \mathbf{y}_i - \mathbf{y} \rangle \right) \\
&= \frac{1}{N} \sum_{i=1}^{N} (\mathbf{y}_i + \frac{\lambda_i}{\rho}) \overset{12:}{=} \frac{1}{N} \sum_{i=1}^{N} \mathbf{y}_i.
\end{aligned}
\tag{17}
$$

Dual variables in line 6 are updated by each agent as follows

$$
\lambda_i \leftarrow \lambda_i + \rho(\mathbf{y}_i - \mathbf{y}), \quad i = 1, \cdots, N.
\tag{18}
$$

Combining (17) and (18), we have $\sum_{i=1}^{N} \lambda_i = 0$.

In line 13 of Algo. 1, after the ADMM process, the server updates global policy parameters, where $\eta \in (0, 1)$ is the step size. The server then broadcasts the updated parameters to all $N$ agents at the next iteration.

In every communication round, agent $i$ only transmits $\mathbf{y}_i$ and $\mathbf{g}_i$, with a communication complexity of $\mathcal{O}(d)$. In contrast, the standard average approach in (12) requires transmitting $\mathbf{H}_i$ and $\mathbf{g}_i$, with a communication complexity of $\mathcal{O}(d^2)$. This efficient communication approach allows second-order methods scalable to large-scale systems.

## 4  Convergence Analysis

In this section, we derive the convergence rate of FedNPG based on ADMM. In order to derive the guarantees, we make the following standard assumptions [1, 6, 23, 31, 51] on policy gradients, second-order matrices, and rewards.

**Assumption 4.1.**

1. The score function is bounded as $\|\nabla_\theta \log \pi_\theta(a \mid s)\| \leq G$, for all $\theta \in \mathbb{R}^d$, $s \in \mathcal{S}$, and $a \in \mathcal{A}$.

2. Policy gradient is $M$-Lipschitz continuous. In other words, $\forall \theta_i, \theta_j \in \mathbb{R}^d$, $s \in \mathcal{S}$, and $a \in \mathcal{A}$, we have
$$
\left\| \nabla_{\theta_i} \log \pi_{\theta_i}(a \mid s) - \nabla_{\theta_j} \log \pi_{\theta_j}(a \mid s) \right\| \leq M \left\| \theta_i - \theta_j \right\|.
\tag{19}
$$

3. The reward function is bounded as $r(s, a) \in [0, R]$, for all $s \in \mathcal{S}$, and $a \in \mathcal{A}$.

**Assumption 4.2.** For all $\theta \in \mathbb{R}^d$, the Fisher information matrix induced by policy $\pi_\theta$ and initial state distribution $\rho$ is positive definite as

$$
F(\theta) = \underset{(s,a) \sim \nu_{\pi_\theta}}{\mathbb{E}} \left[ \nabla_\theta \log \pi_\theta(a \mid s) \nabla_\theta \log \pi_\theta(a \mid s)^\top \right] \succcurlyeq \mu_F \cdot \mathbf{I}
\tag{20}
$$

for some constant $\mu_F > 0$. For any two symmetric matrices with the same dimension, $A \succcurlyeq B$ denotes the eigenvalues of $A - B$ are greater or equal to zero.

**Theorem 4.3.** *For a target error $\epsilon$ of stationary-point convergence, each agent samples $\mathcal{O}(\frac{1}{(1-\gamma)^4 N \epsilon})$ trajectories, and the server obtains the update direction $\mathbf{y}^k$ at each iteration. Choose $\eta = \frac{\mu_F^2}{4G^2(56G^2 + L_J)}$ and*

$$
K = \frac{(J^\star - J(\theta^1))(56G^2 + L_J)^2 16G^2 + 28G^2 \mu_F^3}{(56G^2 + L_J - 56G^2 \mu_F)\mu_F^2 \epsilon} = \mathcal{O}\left( \frac{1}{(1-\gamma)^2 \epsilon} \right).
\tag{21}
$$

*We have:*

$$
\frac{1}{K} \sum_{k=1}^{K} \mathbb{E}\left[ \left\| \nabla J\left(\theta^k\right) \right\|^2 \right] \leq \epsilon.
\tag{22}
$$

**Proof sketch.** The main idea in our proof is to show that the approximation error between the updating direction given by FedNPG-ADMM and NPG geometrically decreases up to some additional term, which depends on the statistical error. This term appears since we do not have access to the exact gradient. However, it decays at a rate proportional to sample size. As a result, FedNPG-ADMM achieves the same convergence rate as NPG as shown by constructing an appropriate Lyapunov function. The complete proof of Theorem 4.3 is given in Appendix A.

By the definition of stationary points, we need to find a parameter $\theta$ such that $\mathbb{E} \left\| \nabla J \left( \theta \right) \right\|^2 \leq \epsilon$, for all $\epsilon > 0$. The result in (22) achieves the stationary-point convergence for policy gradient methods as provided in [51]. Our approach keeps the sample complexity as that in the NPG method [23] and thanks to the federated scenario we consider, enjoys a much lower communication complexity.

Table 1: Complexity comparison in each agent.

|  | NPG [23] | FedNPG | FedNPG-ADMM |
|---|---|---|---|
| Sample complexity | $\mathcal{O}(\frac{1}{(1-\gamma)^6 \epsilon^2})$ | $\mathcal{O}(\frac{1}{(1-\gamma)^6 N \epsilon^2})$ | $\mathcal{O}(\frac{1}{(1-\gamma)^6 N \epsilon^2})$ |
| Communication complexity | - | $\mathcal{O}(\frac{d^2}{(1-\gamma)^2 \epsilon})$ | $\mathcal{O}(\frac{d}{(1-\gamma)^2 \epsilon})$ |

We summarize the complexity improvement in Table 1. Recall in (5) that the estimated gradient $\mathbf{g}$ is an average over collected trajectories. The total trajectories $\sum_{i=1}^{N} |\mathcal{D}_i|$ are collected by $N$ agents equally using the common global policy $\pi_\theta$ at each iteration. Each agent $i$ samples $|\mathcal{D}_i| = \mathcal{O}(\frac{1}{(1-\gamma)^4 N \epsilon})$ at each iteration and enjoys a federated sampling benefit compared to a single agent with $|\mathcal{D}_i| = \mathcal{O}(\frac{1}{(1-\gamma)^4 \epsilon})$. During the whole training process, each agent $i$ has $K|\mathcal{D}_i| = \mathcal{O}(\frac{1}{(1-\gamma)^6 N \epsilon^2})$ sample complexity and $K \cdot 2d = \mathcal{O}(\frac{d}{(1-\gamma)^2 \epsilon})$ communication complexity to achieve a stationary convergence.

## 5 Simulations

**Setup**: We consider three MuJoCo tasks [44] with the MIT License, which have continuous state spaces. Specifically, a Swimmer-v4 task with small state and action spaces, a Hopper-v4 task with middle state and action spaces, and a Humanoid-v4 task with large state and action spaces are considered as described in Table 3. Policies are parameterized by fully connected multi-layer perceptions (MLPs) with settings in Table 4. We follow the practical settings in TRPO with line search [38], and in stable-baselines3 [35] with generalized advantage estimation (0.95) [39] and the Adam optimizer [18] in our implementation. Convergence performances are measured over 10 runs with random seeds from 0 to 9. The solid lines in Figure 2 and 3 are averaged results, and the shadowed areas are confidence intervals with 95% confidence level. We use PyTorch [32] to implement deep neural networks and RL algorithms. The tasks are trained on NVIDIA RTX 3080 GPU with 10 GB of memory.

**Performance metrics**: We consider performance metrics as follows:

1. Communication overhead: the data size transmitted from each agent;

2. Rewards: the average trajectory rewards across the batch collected at each iteration;

3. Convergence: rewards versus iterations during the training process.

We first evaluate the influence of the number of federated agents. In Figure 2, with different numbers of agents, we test the convergence performances of standard average FedNPG in (12) with $\mathcal{O}(d^2)$ communication complexity and FedNPG-ADMM in Algo. 1 with $\mathcal{O}(d)$ communication complexity at each iteration. The x-axis denotes the number of iterations in federated learning. Both algorithms converge faster, have lower variance, and achieve higher final rewards when more agents are engaged to collect trajectories. Compared to the standard average FedNPG, FedNPG-ADMM does not only reduce communication complexity, but also keeps convergence performances. The hyperparameter and MLP settings are described in Table 4. We summarize the final reward results with standard deviations in Table 2. FedNPG-ADMM achieves similar final rewards compared to the standard average FedNPG. It also works with slightly high variance when only one agent is engaged.

In Figure 3, we compare the performances of the standard average FedNPG, FedNPG-ADMM, and first-order FedPPO algorithms. The number of federated agents $N$ is fixed to 8. PPO clipping parameter is set as 0.2. FedNPG methods outperform FedPPO in these tasks with faster convergence and higher final rewards. It is noticeable that FedNPG-ADMM has similar convergence rates and achieves slightly higher final rewards than the standard average FedNPG for the Swimmer-v4 task, while it becomes slightly lower for the Humanoid-v4 task. Generally, there is no significant difference in convergence rates after ADMM approximation. The communication overhead is measured by the number of transmitted parameters with double precision in each agent. FedNPG-ADMM keeps the communication overhead as the first-order methods, while FedNPG has much higher costs. The ADMM method reduces the cost by 4 orders of magnitude in the Swimmer-v4 task and about 6 orders of magnitude in the Humanoid-v4 task.

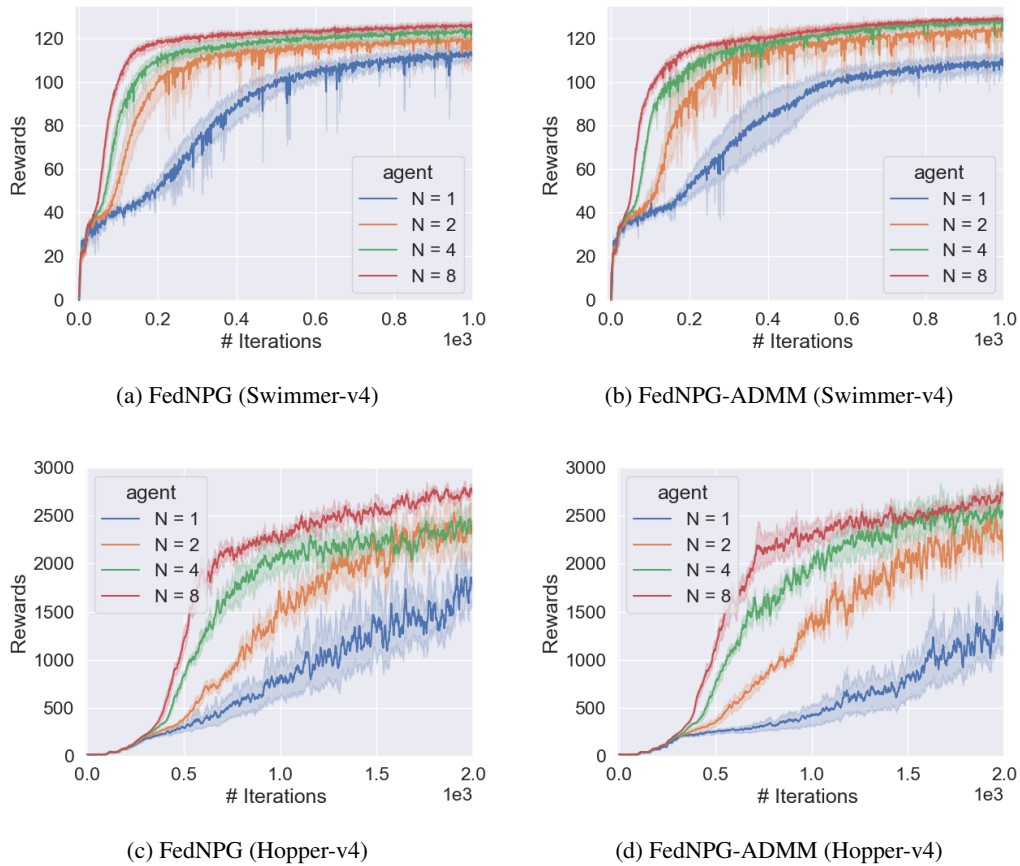

(a) FedNPG (Swimmer-v4)

(b) FedNPG-ADMM (Swimmer-v4)

(c) FedNPG (Hopper-v4)

(d) FedNPG-ADMM (Hopper-v4)

Figure 2: Reward performances of standard average FedNPG and FedNPG-ADMM on MuJoCo tasks, where $N$ is the number of federated agents. **Top:** Swimmer-v4, **Bottom:** Hopper-v4. **Left:** FedNPG with $\mathcal{O}(d^2)$ communication complexity, **Right:** FedNPG-ADMM with $\mathcal{O}(d)$ communication complexity.

Table 2: Final rewards in federated settings.

| # Agents | | 1 | 2 | 4 | 8 |
|---|---|---|---|---|---|
| Swimmer-v4 | FedNPG | $111.9 \pm 5.4$ | $119.5 \pm 3.4$ | $122.1 \pm 3.6$ | $124.8 \pm 2.4$ |
| | FedNPG-ADMM | $109.4 \pm 5.9$ | $123.6 \pm 10.3$ | $127.2 \pm 2.2$ | $128.5 \pm 1.9$ |
| Hopper-v4 | FedNPG | $1644 \pm 396$ | $2468 \pm 426$ | $2458 \pm 171$ | $2736 \pm 158$ |
| | FedNPG-ADMM | $1473 \pm 383$ | $2384 \pm 371$ | $2507 \pm 230$ | $2719 \pm 173$ |

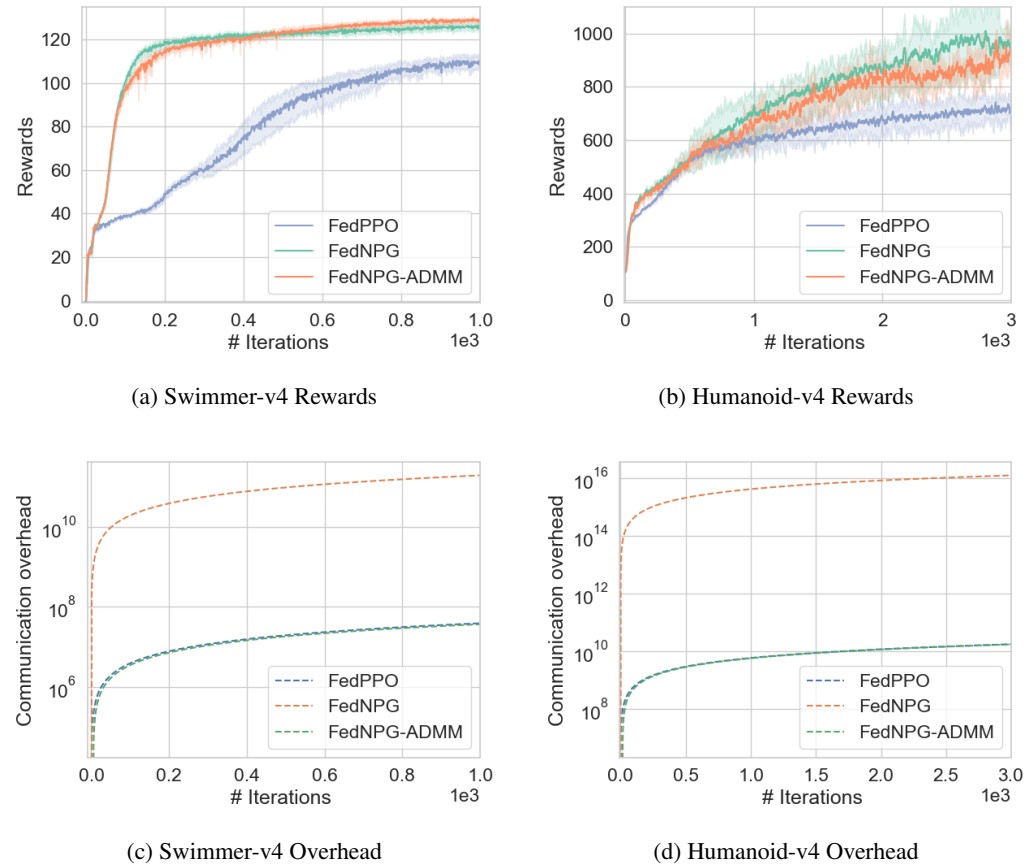

(a) Swimmer-v4 Rewards        (b) Humanoid-v4 Rewards

(c) Swimmer-v4 Overhead        (d) Humanoid-v4 Overhead

Figure 3: Comparisons of FedPPO, standard average FedNPG, and FedNPG-ADMM on MuJoCo tasks, where the number of federated agents $N$ is $8$ and the communication overhead is measured by the transmitted bytes in each agent. **Left:** Swimmer-v4 task, **Right:** Humanoid-v4 task, **Top:** Reward performances, **Bottom:** Communication overhead.

Table 3: Description of the MuJoCo environment.

| Tasks | Agent | Action Dimension | State Dimension |
|---|---|---|---|
| Swimmer-v4 | Three-link swimming robot | 2 | 8 |
| Hopper-v4 | Two-dimensional one-legged robot | 3 | 11 |
| Humanoid-v4 | Three-dimensional bipedal robot | 17 | 376 |

Table 4: Hyperparameter and MLP settings.

| Hyperparameter | Setting | | |
|---|---|---|---|
| Task | Swimmer-v4 | Hopper-v4 | Humanoid-v4 |
| MLP | $64 \times 64$ | $128 \times 128$ | $512 \times 512 \times 512$ |
| Activation function | ReLU | ReLU | ReLU |
| Output function | Tanh | Tanh | Tanh |
| Penalty ($\rho$) | 0.1 | 0.1 | 0.01 |
| Radius ($\delta$) | 0.01 | 0.01 | 0.01 |
| Discount ($\gamma$) | 0.99 | 0.99 | 0.99 |
| Timesteps ($T$) | 2048 | 1024 | 512 |
| Iterations ($K$) | $1 \times 10^3$ | $2 \times 10^3$ | $3 \times 10^3$ |
| Learning rate | $3 \times 10^{-4}$ | $3 \times 10^{-4}$ | $1 \times 10^{-5}$ |

In Figure 4, we test performances with agent selection. In the Swimmer task, we randomly select 75% and 50% of agents in each iteration, and the performances only drop slightly (final rewards drop less than 6%). Thus, our proposed method is robust for agents with disconnection in practice.

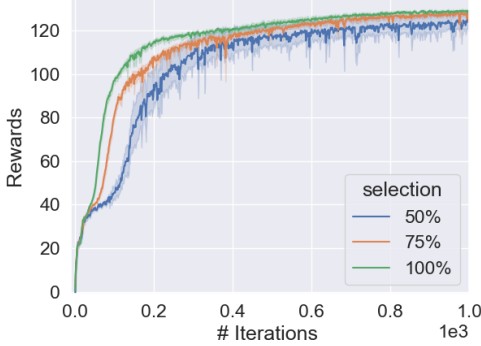

Figure 4: Reward performances of FedNPG-ADMM on the Swimmer-v4 task with agent selection. In each iteration, the server randomly selects 100%, 75%, and 50% of agents for the aggregation.

## 6    Conclusion & Discussion

In this paper, we proposed a novel federated natural policy gradient (NPG) algorithm that estimates global directions using ADMM (FedNPG-ADMM). Our ADMM-based gradient updates significantly reduce communication complexity from $\mathcal{O}(d^2)$ to $\mathcal{O}(d)$, where $d$ is the number of model parameters. It thus enables second-order policy gradient methods to be used for large-scale federated reinforcement learning problems. We proved that our FedNPG-ADMM algorithm achieves stationary convergence without requiring more samples as compared to standard FedNPG. Furthermore, we empirically showed the improved performances of FedNPG-ADMM in the MuJoCo environments as compared to FedNPG and the first order methods.

Overall, our proposed FedNPG-ADMM algorithm provides a scalable and efficient solution for large-scale federated reinforcement learning problems. Our contributions include a novel direction on policy-based FedRL, a new algorithm, reduced communication complexity, and convergence analysis. We believe that our approach can have a significant impact on a wide range of real-world applications, where large-scale distributed reinforcement learning with communication and privacy constraints is critical.

**Limitations:** (1) While the communication complexity improvement is loosely tied to the privacy advantage, exploring deep connections with differential privacy improvement is open. (2) Partial agent participation is not studied under this framework, in which communication complexity can be further reduced. (3) In experiments, it would be more persuasive to extend the number of federated agents to a larger scale.

## Acknowledgments

We thank all reviewers for their invaluable feedback that helped us significantly strengthen the work. C. Brinton acknowledges financial support from NSF CNS-2146171 and DARPA D22AP00168-00. Han Wang and James Anderson acknowledge financial support from NSF grants ECCS-2144634 and ECCS-2231350.

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
