# OpenReview forum: "Improved Communication Efficiency in Federated Natural Policy Gradient via ADMM-based Gradient Updates"
_NeurIPS.cc/2023/Conference — NeurIPS 2023 poster_

### Official Review · Reviewer_ZDkh · 2023-06-27

**Soundness:** 3 good
**Presentation:** 3 good
**Contribution:** 3 good
**Rating:** 6
**Confidence:** 4

**Summary:**

This paper proposes a communication-efficient algorithm, FedNPG-ADMM, for federated natural policy gradient by using a reformulation of quadratic problem. It reduces the communication complexity from $\mathcal{O}(d^2)$ to $\mathcal{O}(d)$. The convergence analysis is provided accordingly.

**Strengths:**

1. The paper is well-written and clearly presented.
2. The reformulation of the original quadratic problem into a distributed manner (eq 14) is a smart move.
3. Using ADMM-based global direction estimation to reduce the communication complexity by $\mathcal{O}(d)$.
4. Stationary convergence in non-convex case is provided, which is the same order as the standard FedNPG.

**Weaknesses:**

As it is in the federated learning (FL) setting, I am curious about main difference between the FL setting considered in this paper and the classic distributed learning.
1. data heterogeneity is one of the key features in FL, but how it impact the model performance does not clearly discussed in the paper, neither in theory or in experiments.
2. To reduce the communication costs, local updates are often used within each agent for FL algorithms. But in the FedNPG-ADMM algorithm proposed in this paper, there is no local update step for each agent. Could local steps be directly applied in the FedNPG-ADMM and further reduce the communication cost while maintaining the same convergence?
Given above, to be precise, the setting for this paper is a distributed learning setting, rather than FL with data heterogeneity and local update.

The major contribution is the communication reduction as claimed in the paper, so I expect the experiments could have more comparison of direct communication costs.
1. Directly comparison in terms of the communication cost should be given. In other words, the results with communication cost as the x-axis are also desired.
2. Appreciate it if other baselines could be included other than FedNPG. I am curious about the actual communication costs among first-order methods, second-order methods (FedNPG), and approximated version of second-order methods (FedNPG-ADMM).

**Questions:**

see above

**Limitations:**

The authors have adequately addressed the limitations.

---

> ### Author Rebuttal · Authors · 2023-08-09
>
> **Q1.a**. Yes, data heterogeneity is a key issue in FL. However, this issue only exists in model aggregation methods (with local updates). As proven in [43], gradient aggregation methods are *immune* to whether collected data is i.i.d., or not. In summary, data heterogeneity will not influence our results, as they are based on gradient aggregation.
>
> **Q1.b**. This can be taken as the same question in Q1.a. FL is a problem and it can be solved in several ways. The key difference between distributed learning and centralized federated learning is the star network [52] in federated learning, where there is a parameter server that collects the updates and broadcasts information to all agents. In the classical distributed learning, clients usually can communicate with each other, while they cannot in the federated setting.
>
> Local updates (FedAvg [24]) are proposed for model aggregation methods.  The argument between local updates and gradient aggregations (called local SGD and minibatch SGD, respectively, in the distributed optimization community) has been going on for years. Proven in [43], gradient aggregation methods are *immune* to whether collected data is i.i.d., or not. [46] also compares the convergence rates between these two paradigms, and the local SGD is actually not better. As the gradient aggregation method is not generally worse than the local update method and is immune to data heterogeneity, we decide to use it. This paper does not aim to argue the differences between them, but to find an efficient way for gradient aggregation in RL.
>
> It is worthwhile to mention that these well-known FL algorithms FedNL [34] and FedPD [62] also use one-step updates. Our algorithm can be extended to multiple local steps. However, multiple local steps may slow down the convergence, as mentioned in the standard FL literature [52]. In this case, it becomes unclear whether FedNPG-ADMM can further reduce the communication cost while maintaining the same convergence. This is a good question and could be future work.
>
> **Q2**. Our original figures follow the conventional settings with the number of iterations as the x-axis. We thank the reviewer for the valuable suggestion. It is indeed better if we have direct communication comparisons as they are the main contributions. Thus, communication comparisons are added in Figure 4 (see added pdf), where communication overhead is measured by the number of transmitted parameters with double precision in each agent. FedNPG-ADMM keeps the communication overhead as the first-order methods, while FedNPG has much larger costs. The ADMM method reduces the cost by $4$ orders of magnitude in the Swimmer-v4 task and about $6$ orders of magnitude in the Humanoid-v4 task.

---

> > ### Comment · Reviewer_ZDkh · 2023-08-14
> >
> > I would like to thank the authors' rebuttal and I will keep the score.

---

### Official Review · Reviewer_pYuC · 2023-07-03

**Soundness:** 2 fair
**Presentation:** 2 fair
**Contribution:** 2 fair
**Rating:** 4
**Confidence:** 4

**Summary:**

The paper studies how to train a global policy using distributed data in reinforcement learning. The authors propose a distributed natural policy gradient method by employing ADMM to approximately compute a natural policy gradient direction. The communication complexity is linear in the dimension of policy parameters. The authors also prove the sublinear iteration complexity for achieving a stationary point. The effectiveness of the proposed method is finally verified in MuJoCo experiments.

**Strengths:**

**originality**

High communication complexity is a bottleneck for implementing many RL algorithms in federated training. I believe the novelty of this work lies in a new application of ADMM to a popular RL algorithm: natural policy gradient. This method achieves better communication complexity than one for the naive method. I am not aware of this method in the literature.

**quality**

- The authors provide the stationary-point analysis of the proposed method in a distributed training setting.

- The authors verify the use of the proposed method in experiments.

**clarity**

The proposed method and theory are clearly explained.

**significance**

- The distributed training algorithm is important for applying RL in distributed systems.

- The stationary-point convergence guarantee is provided.

**Weaknesses:**

- The naive method of fererated natural policy gradient only has quadratic communication complexity with respect to the dimension of policy parameter. This does not seem to be a real bottleneck.

- Directly extending centralized RL algorithms is not very significant, since fault-tolerance and robustness issues are more important to distributed systems.

- It is important to compare communication complexity with the literature, e.g., Fault-Tolerant Federated Reinforcement Learning with Theoretical Guarantee.

- Some assumptions can be strong in practice, e.g., the invertibility of Hessian. It is useful to provide examples when Hessian is invertible.

- The provided convergence property can be arbitrarily sub-optimal, and it is not clear how to check all assumptions. The stationary-point analysis has a gap with the convergence rate analysis of natural policy gradient in the literature.

- The experimental setting is artificial since it is not originally designed for federated learning. There are no comparable baselines in experiments.

**Questions:**


Some questions are raised in Weaknesses. Here are some other questions.

- Can the authors provide robustness or fault-tolerance guarantees?

- Can ADMM-based update be applied to other policy gradient methods? It is useful to discuss generalizability.

- Since the global convergence exists in many policy gradient methods, can the authors strengthen the convergence guarantees?

- Can the authors compare the proposed method with other baselines, e.g., Fault-Tolerant Federated Reinforcement Learning with Theoretical Guarantee?

**Limitations:**

Yes.

---

> ### Author Rebuttal · Authors · 2023-08-09
>
> **W1**. This is actually an important bottleneck for the following reasons. Generally, if one method can reach a large scale, it means that its complexity is at most $O(n)$ [53]. Thus, the $O(n^2)$ complexity in the naive method is not acceptable in large-scale FL. In DRL, a policy is approximated by NNs, and the sizes of NNs are generally large. For toy tasks in MuJoCo [54] or language agents [55], the parameter sizes can reach $10^6$ level, and the sizes of Hessains are ${12}$ orders. In practice, the sizes of NNs in RL are getting larger and larger. With transformers, parameter sizes reach $2.2\times 10^{10}$ [56], and its square is $4.8\times 10^{20}$ ($4$ zettabyte with double-precision). Transmitting these large amounts of data from each agent in each iteration is not realistic.
>
> **W2&Q1**. First, our paper is not a direct extension of central RL. Compared to the central setting, the first critical bottleneck is a limited communication budget in FL [17, 52]. It is necessary to design FedRL that leverages communication costs and learning performances. Our FedNPG-ADMM utilizes the 2nd-order information but only requires its approximation without sending Hessians. As NPG (2nd-order) is more stable and converges faster than PG (1st-order), we rigorously prove the convergence of FedNPG-ADMM.
>
> Second, we agree that fault tolerance is important in distributed systems. However, FedRL is a new and emerging field with a limited number of theoretical publications at this moment. The adaptation of centralized RL algorithms to FL while achieving a balance between communication cost and learning efficiency remains a challenging task, and it is the first step. Our paper thoroughly solved this problem. The fault-tolerance theory is a good direction for the next step. Noticeably, our method is orthogonal to several malicious detection methods, e.g., the attack stealth [57]. To expand our work, we add experiments with agent selection in Figure 5. In the Swimmer task, we randomly select $75\%$ and $50\%$ of agents in each iteration, and the performances only drop slightly (final rewards drop less than $6\%$). Thus, our proposed method is robust for agents with disconnection in practice.
>
> **W3&Q4**. We thank the reviewer for pointing out the valuable work (FedBR) [58], and will add it to our introduction. However, this approach is designed for Byzantine attacks, while we focus on reducing communication costs while maintaining performance.
>
> First, our algorithm converges *faster* than FedBR. FedBR focuses on PG (1st-order). Our work is NPG (2nd-order). [22] thoroughly compare the convergence performances of PG and NPG. If measured in the same metric, our convergence rate is ${O}(1/\epsilon N)$, while ${O}(1/\epsilon^{\frac{5}{3}}N^{\frac{2}{3}})$ in FedBR. Further, our method improves the communication cost in FedBR by $O(d\epsilon^{\frac{2}{3}}/N^{\frac{1}{3}})$. Second, FedBR does not show the sample complexity of FedRL can be scaled linearly by the number of agents, i.e., $\alpha =0$ in Table 1 [58]. In contrast, our paper shows that *the sample complexity is scaled linearly* by $N$, which comes from the benefit of collaboration, and in Figure 2, a thorough comparison is given to support it.
>
> In summary, NPG updates policies according to the Riemannian metric and enjoys stable on-policy explorations. We do not think it is suitable to directly compare them, though.
>
> **W4**. In practical computation, if take an $n\times n$ symmetric matrix with uniformly distributed entries, it will almost certainly be invertible.
>
> The inverse expression is widely used in the original NPG (Theorem 1) [16] and TRPO (Appendix C) [35]. It can be replaced by the MP-pseudoinverse in [2, 35]. It is worth mentioning that in practice, it is expensive to solve $Ax=b$ by computing $A^{-1}$ in both space and computation. A conjugate gradient method is used in DRL to approximate the result [33].
>
> **W5&Q3**. Our assumptions are widely used and verified in previous works. See Q2 in Reviewer1 q8Jc for details.
>
> The convergence of FedNPG-ADMM is fixed-point optimality, and our work is NPG instead of vanilla PG. In FL, even without ADMM, it is still challenging to provide a global guarantee for NPG without additional strong assumptions. The global guarantee of NPG attributes to the gradient domination property [2]. Without it, the convergence can only be assured to local optimums instead of a global one in the nonconvex setting. However, this property is limited to the centralized setting with $N=1$, which is *the reason that lots of global guarantees are established in the centralized setting*. If each function $\\{f_i\\}^N_{i=1}$ satisfies the property, the sum $\sum_{i=1}^N f_i$ might not. We recognize the importance of a global guarantee for the ADMM approach. Finding sufficient (and then necessary) assumptions for FedNPG (even without ADMM) is a direction for future works. Practically, we verify the performances by experiments, and it shows that the FedNPG-ADMM achieves the same convergence as FedNPG.
>
> **W6**. First, we follow the standard setting in FL. In the original FL work [24], the classical image and language tasks are used. In FL research, we usually make classical tasks federated instead of designing new tasks [15, 52].
>
> Second, FedRL is a new field and has not formed experimental baselines. There are datasets published last year for labeled images [59] and health images [60], but they are not for RL and have not formed baselines. We are more than happy to see there are open-source tasks or baselines designed for FL and widely accepted by the community. It would be better if they are designed for FedRL.
>
> **Q2**. Any 2nd-order PG can apply the ADMM method. Regarding other 2nd-order methods, there is a Hessian-aided PG attempt [61], but its conclusion still claims NPG achieves better performances. NPG itself forms a basic approach for several variants [2, 22, 35].

---

> ### Author Response · Authors · 2023-08-18
>
> Dear Reviewer,
>
> We just wanted to check again to see whether our comments have addressed your concerns. We are happy to provide any additional clarifications that may be needed. Thank you again for your time spent reviewing our paper.
>
> Best,
>
> Authors

---

### Official Review · Reviewer_GLbD · 2023-07-05

**Soundness:** 3 good
**Presentation:** 3 good
**Contribution:** 3 good
**Rating:** 7
**Confidence:** 2

**Summary:**

This paper applies the ADMM technique to the Fed-NPG algorithm in reinforcement learning and reduces the communication cost from $O(d^2)$ to $O(d)$ where $d$ is the number of parameters, which nearly maintains the convergence results of Fed-NPG. Empirical results verify the theoretical analysis.

**Strengths:**

The communication cost reduction is impressive since in the federated learning setting, the communication cost is one of the bottlenecks, and the convergence results are nearly the same as Fed-NPG. The idea to combine natural policy gradient with ADMM style methods (or the penalty functions) is interesting (and at least novel to me).

**Weaknesses:**

I am not very familiar with reinforcement learning, and thus may not find potential weaknesses. However, I suggest comparing with the original policy gradient methods in the empirical section and even in the main content (show that Fed-NPG-ADMM converges much fast than the policy gradient variation in federated learning) and plotting all the communication cost for Fed-NPG, Fed-NPG-ADMM, and policy gradients.

**Questions:**

See the weakness section.

---

> ### Author Rebuttal · Authors · 2023-08-09
>
> **Q1**. Thank you for these constructive suggestions.  We will add PG in the main content and communication comparisons are added in Figure 4 (see added pdf), where communication overhead is measured by the number of transmitted parameters with double precision in each agent. FedNPG-ADMM keeps the communication overhead as the first-order methods, while FedNPG has much higher costs. The ADMM method reduces the cost by $4$ orders of magnitude in the Swimmer-v4 task and about $6$ orders of magnitude in the Humanoid-v4 task.

---

> > ### Comment · Reviewer_GLbD · 2023-08-14
> >
> > Thanks for your response. Since I am not very familiar with the RL background, I will maintain my score and confidence.

---

### Official Review · Reviewer_q8Jc · 2023-07-23

**Soundness:** 3 good
**Presentation:** 3 good
**Contribution:** 2 fair
**Rating:** 6
**Confidence:** 4

**Summary:**

The work proposed a new algorithm for federated policy optimization. The proposed work uses prima-dual update to replace the primal update of FedNPG, so that the communication cost reduces from $d^2$ to $d$. The proposed work enjoys same rate of convergence as FedNPG under certain assumptions, and numerical experiments were proposed to show the efficacy of the algorithm.

**Strengths:**

The paper is well-rounded and the presentation is clear. The proposed method clearly enjoys the supremacy in terms of communication error. The numerical experiments were proposed to show the efficacy of the algorithm.

**Weaknesses:**

(Please reply to the Questions section directly) The motivation of the proposed algorithm seems a bit less significant. Moreover, the convergence theory requires policy gradient bounded, which seems quite strong. In the numerical experiments, more results could be included to further show the efficiency of the proposed algorithm.

**Questions:**

Here are the my questions and suggestions:

1. Concerns on the motivations: the idea of combining reinforcement learning and federated learning is interesting, which is also largely un-explored in the literature. It is good to see the exploration on this direction. An intriguing question is that can we consider existing works on reformulating the TRPO, such as the KL-penalitzed objective in [1], to federated setting? Since if we get rid of the constraint, we don’t need to communicate the Hessian in the first place (and FedAvg-type algorithm may also enjoys similar rate of convergence in theory). In summary, the authors propose to study the policy optimization in federated setting, but eventually ended up at a less significant improvement over FedNPG (and this improvement comes from essentially the constrained federated learning, not from RL or policy optimization), which makes me feel less confident about the contribution in the RL field;

2. In terms of the assumptions for convergence analysis, the authors proposed bounded policy gradient and Lipschitz continuous (Assumption 4.1) and claimed that they are fairly standard in the existing literature. I’m actually not very sure about the boundedness of the policy gradient. Could the authors just include the specific assumptions in the existing literatures? Same question for Assumption 4.2.

3. The penalty parameter $\rho$ is the penalty to make sure that each local $y_i$ consensus with the global $y$. In the experiment $\rho$ is simply a small number as 0.1 or 0.01. My concern is that during the update, is $y_i - y\rightarrow 0$ true or not? The authors could also plot the consensus errors.

4. In the experiments, for example Figure 2 and 3, FedNPG-ADMM seems not better than FedNPG in terms of the rewards. It would be interesting to see why this is the case. I know that the main claim of FedNPG-ADMM is its commutation efficiency, and I think the authors could also plot the curves wrt the CPU times.

References:

[1] Schulman, John, et al. "Proximal policy optimization algorithms." arXiv preprint arXiv:1707.06347 (2017).

**Limitations:**

The limitation is well stated in weakness and question sections. The authors also include the discussions in limitations at the end of the work.

I’m not aware of any potential negative social impact of this work.

---

> ### Author Rebuttal · Authors · 2023-08-09
>
> **Q1**. This is a thoughtful suggestion. We had the same thought at the beginning. However, local updates in the FedAvg-type might not bring advantages compared to gradient aggregation [46, 43]. On the other hand, unlike supervised learning, local updates in RL bring different **local policies**. We do not think it is a good idea to use **on-policy** algorithms for samples collected by different local policies.
>
> Second, although the soft penalty approach has simpler forms, it might bring worse performances than the constraint methods, e.g., on the Swimmer [37] and Humanoid [10] tasks in MuJoCo environments. With a hard constraint, the NPG enjoys stable training performances and has solid theoretical guarantees [2, 22]. In Figure 3, our experiments also verify that FedPPO has worse performances.
>
> Third, as for the first-order PG, [22] thoroughly compares the convergence performances of PG and NPG. Furthermore, recent works have demonstrated that the convergence guarantees of NPG with KL divergence constraints are superior to those of PG [20, 22], motivating closer inspection for practical use.
>
> In summary, this work makes the second-order methods have the same communication complexities as the first-order methods per iteration, and achieve stable and higher rewards in the federated setting.
>
> **Q2**. The bounded score function assumption is used in Assumption 4.1 [47], Assumption 5.1 [48], Assumption 4.1 [45], Assumption 4.2 [22], and Assumption 3 [49]. The positive definite assumption is used in Assumption 2.1 [22]. The continuity assumption is used in Assumption 4.1 [47], Assumption 5.1 [48], Assumption 4.1 [45], Assumption 4.2 [22], Assumption 2 [49], and Assumption 6.4 [2] (and the following lemmas). The above assumptions are verified for simple policy parameterizations such as Gaussian policies [47, 50, 51].
>
> **Q3**. True. We test the averaged differences $\frac{\sum_{i=1}^{N}\lVert y_i - y \rVert^2}{N}$ with respect to the number of iterations. The differences gradually decrease to $0$. Over the course of training, policies are more deterministic and less random. It is not surprising that $y_i - y \rightarrow 0$ as policies become stable.
>
> **Q4**. In the experiments, we run each task ten times with random seeds from $0$ to $9$. Sometimes FedNPG-ADMM achieves higher rewards than FedNPG, while the averaged results slightly drop in the Humanoid task. As the ADMM approach is, in fact, an approximation of the exact aggregation, it is not surprising that the ADMM approach sometimes drops a little. In any way, they generally have the same convergence rate and similar performances in these tasks. We thank the reviewer for the computing time suggestion. However, communication time dominates the cost in federated learning [3, 15, 52]. To make the comparisons more clear, with reference to the comments of two other reviewers, we add the communication overhead of each method in Figure 4. The communication time (byte transmission) is reduced by $4$ orders of magnitude in the Swimmer-v4 task and about $6$ orders of magnitude in the Humanoid-v4 task.

---

> > ### Comment · Reviewer_q8Jc · 2023-08-13
> >
> > Thank you for your detailed responses to my comments and questions. I believe that paper is useful contribution to the field and the rebuttal addressed my biggest concern. I appreciate it if the authors could modify the paper accordingly.
> >
> > I'll raise my evaluation to 6.

---

> > > ### Author Response · Authors · 2023-08-14
> > > **Thank you**
> > >
> > > Thank you for your consideration. We will be sure to modify the paper accordingly.

---

### Author Rebuttal · Authors · 2023-08-09

Dear Area Chairs and Reviewers,

We appreciate your organization and valuable feedback.

In the one-page pdf, we add communication costs in Figure 4 and agent selection in Figure 5.

In the rebuttals, references [1-45] are from the main paper, and references [46-62] are listed as follows:

---
[46] Woodworth, B., Patel, K.K., Stich, S., Dai, Z., Bullins, B., Mcmahan, B., Shamir, O., Srebro, N.: Is local SGD better than minibatch SGD? ICML (2020)

[47] Papini, M., Binaghi, D., Canonaco, G., Pirotta, M., Restelli, M.: Stochastic variance-reduced policy gradient. ICML (2018)

[48] Xu, P., Gao, F., Gu, Q.: An improved convergence analysis of stochastic variance-reduced policy gradient. In: Proceedings of The 35th Uncertainty in Artificial Intelligence Conference (2020)

[49] Ding, D., Zhang, K., Basar, T., Jovanovic, M.: Natural policy gradient primal-dual method for constrained Markov decision processes. NeurIPS (2020)

[50] Cortes, C., Mansour, Y., Mohri, M.: Learning bounds for importance weighting. NeurIPS (2010)

[51] Pirotta, M., Restelli, M., Bascetta, L.: Adaptive step-size for policy gradient methods. NeurIPS (2013)

[52] Li, T., et al: Federated learning: Challenges, methods, and future directions. IEEE signal processing magazine 37(3), 50–60 (2020)

[53] Bottou, L., et al: Optimization methods for large-scale machine learning. SIAM Review (2018)

[54] Liu, X.Y., et al: Stationary deep reinforcement learning with quantum k-spin Hamiltonian regularization. ICLR Workshop on Physics for Machine Learning (2023)

[55] Zhong, V., et al: Improving policy learning via language dynamics distillation. NeurIPS (2022)

[56] Dehghani, M., et al: Scaling vision transformers to 22 billion parameters. ICML (2023)

[57] Bhagoji, A.N., et al: Analyzing federated learning through an adversarial lens. ICML (2019)

[58] Fan, X., et al: Fault-tolerant federated reinforcement learning with theoretical guarantee. NeurIPS (2021)

[59] Song, C., et al: Federated learning annotated image repository. NeurIPS (2022)

[60] Ogier du Terrail, J., et al: Datasets and benchmarks for cross-silo federated learning in realistic healthcare settings. NeurIPS (2022)

[61] Shen, Z., et al: Hessian aided policy gradient. ICML (2019)

[62] Zhang, X., Hong, M., Dhople, S., Yin, W., Liu, Y.: FedPD: A federated learning framework with adaptivity to non-iid data. IEEE Transactions on Signal Processing 69, 6055–6070 (2021)

---

### Decision · Program_Chairs · 2023-09-21

**Decision:**

Accept (poster)

**Comment:**

This paper studies a communication efficient approach for federated reinforcement learning. Overall the reviewers are impressed by the communication savings, but are requesting more extensive numerical results to demonstrate the benefits. Further, the work is a bit incremental since it is a direct extension to the centralized case, using ADMM, and it is sort of expected that such approach will work.